# Bio-Efficacy of Chrysoeriol7, a Natural Chemical and Repellent, against Brown Planthopper in Rice

**DOI:** 10.3390/ijms23031540

**Published:** 2022-01-28

**Authors:** Eun-Gyeong Kim, Sopheap Yun, Jae-Ryoung Park, Yoon-Hee Jang, Muhammad Farooq, Byoung-Ju Yun, Kyung-Min Kim

**Affiliations:** 1Division of Plant Biosciences, School of Applied Biosciences, College of Agriculture and Life Sciences, Kyungpook National University, Daegu 41566, Korea; dkqkxk632@naver.com (E.-G.K.); icd92@naver.com (J.-R.P.); uniunnie@naver.com (Y.-H.J.); mfarooqsr@gmail.com (M.F.); 2Graduate School of Science, Royal University of Phnom Penh, Sangkat Teuk Laak 1, Russian Federation Boulevard, Toul Kork, Phnom Penh 12101, Cambodia; yun.sopheap@rupp.edu.kh; 3Department of Crop Breeding, National Institute of Crop Science, Rural Development Administration, Wanju 55365, Korea; 4School of Electronics Engineering, College of IT Engineering, Kyungpook National University, 80, Daehak-ro, Buk-gu, Daegu 41566, Korea

**Keywords:** brown planthopper (*Nilaparvata lugens* Stal.), rice, secondary metabolite, chrysoeriol7, environmentally friendly pesticides

## Abstract

Brown planthopper (BPH, *Nilaparvata lugens* Stal.) is the most damaging rice pest affecting stable rice yields worldwide. Currently, methods for controlling BPH include breeding a BPH-resistant cultivar and using synthetic pesticides. Nevertheless, the continuous cultivation of resistant cultivars allows for the emergence of various resistant races, and the use of synthetic pesticides can induce environmental pollution as well as the emergence of unpredictable new pest species. As plants cannot migrate to other locations on their own to combat various stresses, the production of secondary metabolites allows plants to protect themselves from stress and tolerate their reproduction. Pesticides using natural products are currently being developed to prevent environmental pollution and ecosystem disturbance caused by synthetic pesticides. In this study, after BPH infection in rice, chrysoeriol7 (C7), a secondary metabolite that induces resistance against BPH, was assessed. After C7 treatment and BPH infection, relative expression levels of the flavonoid-related genes were elevated, suggesting that in plants subjected to BPH, compounds related to flavonoids, among the secondary metabolites, play an important role in inducing resistance. The plant-derived natural compound chrysoeriol7 can potentially thus be used to develop environmentally friendly pesticides. The suggested control of BPH can be effectively used to alleviate concerns regarding environmental pollution and to construct a relatively safe rice breeding environment.

## 1. Introduction

The brown planthopper (*Nilaparvata lugens* Stal., BPH) (Homoptera: Delphacidae) is a rice (*Oryza sativa* L.) pest. Homoptera causes several forms of damage, such as leaf blights and viral infections while sucking rice stems, and has provided a direct cause for the reduction in rice yield in Asia for decades [1]. In particular, among *Homoptera*, BPH has the greatest contribution the decrease in yield and grain quality [2,3]. BPH sucks leaf sheath, reduces nutrients in plants, destroys tissues through its spawn, impedes nutrient migration, and causes plant colonization. Depending on the growing stage of rice, the resistance against BPH is at different levels, reducing the tiller number and 1000-grain weight, and having a negative effect on yield and grain quality [4,5]. The control of BPH is predominantly achieved by means of synthetic pesticides [6]. BPH inhabits rice stems and has a high replication rate [7]; its control using chemical pesticides is thus relatively ineffective [8]. The continuous use and abuse of synthetic pesticides has led to devastating consequences, such as the pollution of agriculture and of the environment [9], toxicity to humans and livestock [10], and a reduction in their effectiveness due to the outbreak of synthetic pesticide-resistant pathogens [11]. In fact, research has been reported in which speciation was shown to cause resistance to synthetic pesticides, due to the control of BPH with synthetic pesticides. The main components of these pesticides include organophosphates, carbamates, pyrethroids, neonicotinoids, insect growth regulators, and henylpyrazole [8]. However, pesticides using a natural substance can solve such problems [12].

To this end, an alternative strategy that can minimize the damage caused by BPH but provides a solution regarding the chemical pesticides issues is vital [13]. In addition, there is an urgent need for the development and dissemination of bio-pesticides that are able to control plant pathogens using plant-derived natural products without having a negative effect on the ecosystem, such as residual toxicity and environmental pollution [14].

Plants can survive herbivore attacks through a variety of biochemical changes [15]. As plants cannot migrate to other places, various defense systems have been developed in order for them to reproduce and protect themselves from threats [16]. Plants have important defense systems such as antixenosis, antibiosis, and tolerance to improve their tolerance of abiotic/biotic stress [17]. Plant defense systems have been reported to be effective against abiotic stress such as drought, heat, cold, and salinity, as well as insect and virus suppression [18]. The following is an example of a plant showing resistance to abiotic/biotic stress. The synthesis level of salicylic acid (SA) was increased due to infection with potato virus Y (PVY). The SA suppresses virus division, impedes the virus from moving into cells, and has resistance [19]. Antixenosis prevents insects from accessing the host plant, sucking, spawning, and colonizing the plant [20]. Jasmonic acid (JA) is synthesized by infection with *Bemisia tabaci* (Gennadius) and induces necrosis of the infected cells and has resistance [21]. Plants possess a sophisticated and interrelated network of defense strategies to avoid or tolerate abiotic/biotic stress. Physical barriers are the first defense in plants against external stress. Some morphological structures such as thick cuticles and the development of wax layers [22], biologically active tissues such as trichomes [23], and the synthesis of secondary metabolites can act as barriers when plants are faced with external stress [24]. For example, *Arabidopsis thaliana* increases the leaf trichome density when infected by insects, inducing JA synthesis, indicating resistance [25]. By increasing flavonoid accumulation in the plants, the rate of larval infections such as *Anticarsia gemmatalis* (Hübner) (Lepidoptera: Noctuidae) is reduced [26]. In addition to the synthesis of phytoalexin [27], plants can also synthesize various substances such as secondary metabolites to protect themselves [28]. Chalcone synthase, for instance, is an important enzyme involved in flavonoid biosynthesis and promotes salicylic acid formation in order to resist stress such as bacterial and fungal infections in rice [29]. A large amount of kaempferol-3,7-dirhamnoside, which is a flavonoid family, is also synthesized in *Arabidopsis* due to the overexpression of *MYB75* transcription factor, inducing resistance to *Pieris brassicae* (Lepidoptera: Pieridae) [30]. Similarly, some flavonoids respond to insects [31,32], such as the vitexin protein, which inhibits the growth of *Spodoptera litura* (Lepidoptera: Noctuidae) larvae [33], the schaftosides protein, which suppresses the growth of nematodes (*Meloidogyne incognita*) [34], and chrysoeriol, which suppresses the growth of BPH [35].

Chrysoeriol is a flavonoid commonly found in crops such as rice, barley (*Hordeum vulgare*), wheat (*Triticum aestivum*), millet (*Sorghum bicolor*), corn (*Zea mays*), and multiple land plants. Crysoeriol and its derivative, known as O-methylluteolin, have roles as allelochemicals [36] and insect repellents [37]. These secondary metabolites may also present effectiveness within the same species of pests [38]. BPH, for instance, as well as another species, the whitebacked planthopper (WBPH), are resistant to S-linalool [3,39]. The aim of this study was thus to identify whether crysoeriol7 (C7) [40], for which WBPH has been shown to develop resistance, can effectively repel BPH.

In the current study, we assessed the effectiveness of C7 against BPH resistance and analyzed the relative expression level of genes involved in flavonoid biosynthesis in rice with a composition for controlling rice pests containing C7. The aim of this research was to identify new plant-derived extracts for the effective control of BPH. The BPH-control chemical demonstrated by this research was a plant-derived environmentally friendly compound and can be used as a substitute for synthetic pesticides. Therefore, this can lead to the development of a safe rice breeding environment by using a BPH control method that is harmless to nature and humans.

## 2. Results

### 2.1. Analysis of Phenotypes in Rice after Infection with BPH

To identify BPH resistance, plant heights were measured on days 1, 2, and 3 after BPH infection to the Samgang, Nagdong, TN1, SNDH29, SNDH30, and SNDH11 (Figure 1 and Figure 2). Figure 1a shows representative plants of each group in the repeated experiment. Samgang, SNDH29, and SNDH30 were resistant populations to BPH. Nagdong, TN1, and SNDH11 were susceptible populations to BPH. BPH-resistant populations demonstrated no significant differences between the control and BPH infection group. However, the BPH-susceptible population experienced a negative effect on growth compared to the control. The plant heights of the BPH-susceptible population were short from 2 days after BPH infection. In the BPH-susceptible population, the degree of growth disorder due to BPH infection increased over time.

### 2.2. Analysis of Concentration of C7 after BPH Infection

To identify that C7 is involved in BPH resistance, the concentration of C7 after BPH infection was investigated in Samgang, a cultivar resistant to BPH, and Nagdong, a cultivar susceptible to BPH (Figure 3). In Samgang, the C7 concentration of the BPH-infected population was 15.30 ± 1.01, the uninfected population was 12.32 ± 0.85, and the C7 concentration rate of increase (%) was 24.24 ± 1.79 (Table 1). In Nagdong, the C7 concentration of the BPH-infected population was 12.48 ± 0.96, the uninfected population was 11.54 ± 1.21, and the C7 concentration rate of increase (%) was 8.4 1± 6.41. The BPH-infected population had a greater C7 concentration than the uninfected population in Samgang, and Samgang synthesized more C7 than Nagdong (*p* < 0.05).

### 2.3. Assessment of C7 Efficacy against BPH

To identify the effectiveness of isolated C7 against BPH, Samgang, Nagdong, and TN1 were treated with C7, then infected with BPH, and the bio-scoring value and chlorophyll contents were measured 1, 2, and 3 weeks after infection (Figure 4). The bio-scoring value of the C7 + BPH infection group was significantly lower than that of the BPH infection group. The reduction rate of the bio-scoring value was larger in Nagdong and TN1 than in Samgang. There was no difference in Samgang’s chlorophyll content with or without C7 treatment. Nagdong and TN1 had increased chlorophyll contents when treated with C7. In Samgang, Nagdong, TN1, SNDH29, SNDH30, and SNDH11, the C7 + BPH infection group had fewer growth disorders than the BPH infection group (Figure 5). The BPH-resistant population showed no differences in plant height with or without C7 treatment. The BPH-susceptible population had an increased growth rate when treated with C7 compared to controls (Figure 6). Nagdong, TN1, and SNDH11 were less affected by growth disorders in the BPH-infected population after C7 treatment than in the BPH-infected population without C7 treatment.

### 2.4. Flavonoid and Plant Resistant Gene Expression Levels Analysis in BPH-Infected Rice

The expression levels of flavonoid and plant resistant genes were compared in Samgang, Nagdong, TN1, BPH-resistant SNDH29, SNDH30, and BPH-susceptible SNDH11 over time after C7 treatment and BPH infection (C7 + BPH) (Figure 7). Relative expression levels were measured for *flavanone 3-hydroxylase* (*F3H*) and *choristmate mutase* (*CM*), which are known as flavonoid biosynthesis-related genes, as well as for *non-expressor of pathogenesis-related genes1* (*NPR1*) and *WRKY45,* which are known as plant resistant genes. Expression levels of *OsF3H*, *OsCM*, and *OsWRKY45* were significantly lower in plants treated with C7 and infected with BPH compared to control plants infected with BPH (*p* < 0.01) for 1, 2, and 3 days. The expression levels of *OsNPR1* did not show any significant difference between the plants treated with C7 and infected with BPH and the plants only infected with BPH (control).

## 3. Discussion

When the rice was infected with BPH in this search, the level of C7 synthesis in the BPH-resistant population was higher than that in the BPH-susceptible population. After the infection of Samgang, Nagdong, TN1, SNDH29, SNDH30, and SNDH11 with BPH, plant heights were measured to identify the level of effectiveness against BPH. Bio-scoring is a method of classifying the phenotype of each rice cultivars after BPH infection into resistant or susceptible according to the data presented by The International Rice Research Institute (IRRI). Additionally, plant height and chlorophyll contents are the most useful data to analyze resistance to biotic stress, such as WBPH and BPH. In particular, plant height is the best agricultural trait for classifying resistant and susceptible populations after BPH infection. Plant height and chlorophyll contents are agricultural traits that can most easily classify the response to biotic stress as a phenotype and a resistant or susceptible population. Plant height and chlorophyll contents are the characteristics that show the greatest change after BPH infection. Therefore, in many studies, the resistant population and the susceptible population were classified using the plant height and chlorophyll contents [41,42,43]. The plant heights were investigated on days 1–3, and the growth rate was found to be reduced compared to control plants that were not infected with BPH. When a plant is attacked by a herbivorous insect, its growth rate was reduced [44]. This was responsible for BPH-infected Samgang, Nagdong, TN1, SNDH29, SNDH30, and SNDH11 populations having shorter plant heights compared to control plants. The bio-scoring value of the C7 + BPH infection group was lower than that of the BPH-infected group and the chlorophyll content was higher than that of the BPH-infected group. The C7 + BPH infection group had a greater plant height, a lower bio-scoring value, and higher chlorophyll content, and had fewer growth disorders than the BPH infection group, suggesting that C7 is resistant against BPH.

C7 is a compound containing a phenol group, as a compound of the flavonoid family [45,46]. Phenol is volatile, colorless and as an aromatic compound, it has a peculiar scent [47]. Phenol is used to produce preservatives, disinfectants, synthetic resins, dyes, explosives, etc. This type of rice’s volatile compounds attract BPH’s natural enemies and induce indirect resistance [48]. Volatiles induced after BPH sucking attract, for instance, the parasite insect *Anagrus nilaparvatae* and the predator *Cyrtorbinus lividipennis* to defend them from BPH [49]. Nevertheless, the degree of attraction depends on the genotype of the rice [50]. Recent studies have reported that S-linalool, a rice volatile compound induced upon BPH sucking, attracts predators and parasites and repels BPH [39]. The green leaf volatiles (GLV) released by the BPH attack also act as repellents on BPH [51], and it is thus hypothesized that C7, which contains an aromatic compound emitting a volatile and unique aroma, could also infer resistance to BPH in rice by spreading its aroma, which repels BPH.

Flavonoids are a type of plant and fungal secondary metabolite, with polyphenol compounds known as plant defense compounds, and are contained in the majority of plants [52]. In the current work, the relative expression levels of *OsF3H* [53], *OsCM* [54], *OsWRKY45* [55], and *OsNPR1* [56], which are known as plant resistance genes, were compared, and it was identified whether they are also effective against BPH. After infection with BPH, the relative expression levels of *OsF3H*, *OsCM*, and *OsWRKY45* in the Samgang and SNDH29 were relatively higher than in the Nagdong and SNDH11. *OsF3H* and *OsCM* presented higher relative expression levels compared to *OsWRKY45* and *OsNPR1*. Furthermore, *OsF3H* and *OsCM* are genes involved in flavonoid biosynthesis, and their relative expression levels were found to be increased after BPH infection, suggesting that flavonoids are involved in the secondary metabolites synthesized after BPH attacks. The relative expression levels of *OsF3H*, *OsCM*, *OsWRKY45*, and *OsNPR1* were lower in the group infected with BPH after treatment with C7 compared to the group infected with BPH and no C7 treatment. It was thus suggested that C7, acting as a repellent for BPH, infers resistance to BPH in rice [57].

The results of this study demonstrate that C7, a plant defense metabolite that infers resistance to BPH, is an eco-friendly and safe compound that can be used effectively as an alternative to synthetic pesticides, thus contributing to the alleviation of environmental pollution concerns.

## 4. Materials and Methods

### 4.1. Plant Material and Field Design

In this study, 113 double haploid populations, developed by anther cultures of F_1_ obtained by crossing the BPH-resistant cultivar Samgang and the BPH-susceptible cultivars Nagdong and TN1, were used as the experimental rice. The SNDH group was a high generation group used as an intermediate model that has seen generation advancements in the field of Kyungpook National University (Daegu, Korea) since 2010. Among 113 SNDH populations, SNDH29, which is a BPH-resistant population, and SNDH11, which is a BPH-susceptible population, were used as experimental rice for evaluation of the effectiveness of C7 on BPH [54]. Seeds were disinfected using a seed disinfectant (Spotak, Samgong, Korea) and cultured in the dark at 25 °C for 4 days. The disinfected seeds were sown in the field of Kyungpook National University on 15 April 2021 and transplanted at a planting distance of 30 × 15 cm on 20 May 2021. Each population was transplanted in one row, and each row contained 23 plants. The amount of fertilizer used was N-P_2_O_5_-K_2_O = 9–4.5–5.7 kg/10a, and rice was cultivated based on the standards of the Agricultural Science and Technology Research Institute of the Rural Development Administration.

### 4.2. BPH Rearing

Fifty male and female BPHs were obtained from the Agricultural Science and Technology Research Standards of the Rural Development Administration. BPHs were bred in a special cage (50 × 50 × 40 cm) made of acrylic board, and a 100 µm mesh net was used for ventilation at the back of the cage. The breeding was maintained at a temperature of 27 °C, a humidity of 60–70%, and a luminous intensity of 2000 lux in 16 h cycles. BPHs were fed using Chucheong in the seedling stage, and feed was renewed every two weeks.

### 4.3. Evaluation of BPH Resistance in SNDH

In order to confirm the resistance of the selected SNDH population to BPH, the plant heights in the BPH-infected and the control groups were measured. Samgang, Nagdong, TN1, SNDH29, SNDH30, and SNDH11 were infected with 10–15 BPH (2nd–3rd instar), and plant heights were measured after 1, 2, and 3 days of infection (Appendix A Appendix A). The plant height was measured from the bottom of the stem to the edges of the flag leaf. BPH infected 5 plants in each population, and the height was measured for each plant. The plant height was measured five times for each population, and the mean and standard deviation values were calculated.

### 4.4. Isolation of C7 in Rice

After sampling, the leaves of BPH-infected Samgang, Nagdong, TN1, SNDH29, SNDH30, and SNDH11 were ground in a mortar using liquid nitrogen. After adding 10 g of the plant sample obtained previously to the 1000 mL Erlenmeyer flask and 350 mL of 70% methanol, the plants were crossed by shaking at 20 °C. and 130 rpm and reacted overnight. The sample was then filtered twice using filter paper (Whatman Qualitative Filter Paper, Grade 2, UK). The same volume of n-hexane as the filtered sample was added to the separatory funnel, then shaken vigorously and mixed thoroughly. Then, when the layers of the sample were completely separated, the bottom layer of the sample solution was collected, and this sample was concentrated using a rotary evaporator. This concentration was cultured in a water bath at 30 °C, at 6–8 rpm, and used cooling water at −3 to 5 °C. The concentrated sample solution was aliquoted on a glass column (10 × 250 mm in diameter) containing 125 g of Silicagel60 (70–230 mesh, DUKSAN, Ansan, Korea). Then, 20% methanol was aliquoted in 1 mL until the sample solution passed through the edge of the glass column with silica gel filtration. The sample solution filtered through silica gel was collected in a 1.5 mL tube and then dried using a heat blot at 50 °C overnight. Additionally, the expression level of C7 in the sample dried using TLC silica gel 60F254 plates (Merck, KgaA, Darmstadt, Germany) with a ratio of chloroform:methanol:1-butanol:water = 4:5:6:4 was confirmed. After separating, only C7 on TLC plates was isolated using a 4% methane solution.

### 4.5. Evaluation of C7 Efficacy against BPH

To validate that C7 is effective in controlling BPH, Samgang, Nagdong, TN1, SNDH29, SNDH30, and SNDH11 in the seedling stage were treated with C7 and infected with 100 BPH (2nd–3rd). C7 was received at the Plant Molecular Breeding Lab at Kyungpook National University [40]. After spraying the seedling stages of Samgang, Nagdong, TN1, SNDH29, SNDH30, and SNDH11 with 1000 ppm C7, bio-scoring, chlorophyll contents, and plant height were measured. The correlation between phenotype and the C7 was analyzed. The BPH effectiveness of C7 was evaluated based on the differences in plant heights. The plant heights were measured from the bottom of the stem to the end of the flag leaf. To investigate C7’s efficacy in conferring BPH resistance to the cultivar, bio-scoring values were evaluated at 1, 2, and 3 weeks after BPH infection of Samgang, Nagdong, and TN1. The bio-scoring value was assigned based on plant damage evaluation by BPH infection [58]. The bio-scoring value was assigned 0 points if the plant was not damaged, 1 point if there was some damage, 3 points if the leaves were slightly underdeveloped, 5 points if more than half of the leaves were underdeveloped, 7 points if more than half of the plants had died, and 9 points when the plant ultimately died. Chlorophyll contents were investigated using the SPAD-502 Plus (Konica Minolta Optics, Japan) at 3 weeks after C7 + BPH infection. All data were measured five times for each population, and the mean and standard deviation values were calculated.

### 4.6. Comparison of the Expression Levels of Plant Resistance-Related Genes

Samgang, Nagdong, SNDH29, and SNDH11 were infected with BPH, RNA was extracted from the leaves on days 0, 1, 2, and 3 after BPH infection, and the relative expression level of plant resistance-related genes was analyzed. Total RNA was extracted from the leaves using the RNeasy plant mini kit (QIAGEN, Hilden, Germany). Then, 1 µg of RNA was used as a template for cDNA synthesis, and cDNA was synthesized using the qPCRBIO cDNA Synthesis kit (PCRBIOSYSTEM, Wayne, PA, USA). Quantitative real-time PCR (qPCR) was analyzed with the Eco Real-Time PCR System using plant resistance-related gene-specific primers. Reaction solutions for qPCR included 10 μL of 2× Real-time PCR Master Mix (BioFACT, Daejeon, Korea), 2 μL of cDNA, 1 μL of forward primer (10 pmol/μL), and 1μL of reverse primer (10 pmol/μL). ddH_2_O was then added to reach a final volume of 20 μL. *OsActin* was used as a housekeeping gene. Each reaction was repeated three times to calculate mean and standard deviation values.

### 4.7. Statistical Analysis

Statistical software SPSS software (IMMSPSS Statistics, version 22, IBMSPSS Statistics, version 22, Redmond, WC, USA) was used to calculate significant differences (*p* < 0.05) using ordinary one-way ANOVA. One-way ANOVA was used to determine statistically significant differences between the mean values of 3 groups (*p* < 0.05), followed by Duncan’s multiple range test (DMRT). All experimental data were replicated five times.

## 5. Conclusions

BPH seriously damages rice growth and yield. In this research, rice populations were infected with BPH, and the plant height, bio-scoring, and chlorophyll contents were measured to identify populations resistant to BPH. Compared to the group infected with BPH, in the group treated with C7 and then infected with BPH, it was revealed that plant growth increased with time and C7 demonstrated effectiveness against BPH. In addition, after infection with BPH, the relative expression levels of genes involved in flavonoid biosynthesis was increased, and as a result, it was demonstrated that flavonoids are involved in the secondary metabolites synthesized as protective substances against BPH attacks in rice. The current study also suggested that C7 is a natural flavonoid compound derived from plants and is efficient in BPH control. The identified BPH-effective compound C7 can be potentially used to develop environmentally friendly pesticides, and its use is expected to provide solutions with regard to environmental pollution.

## Figures and Tables

**Figure 1 ijms-23-01540-f001:**
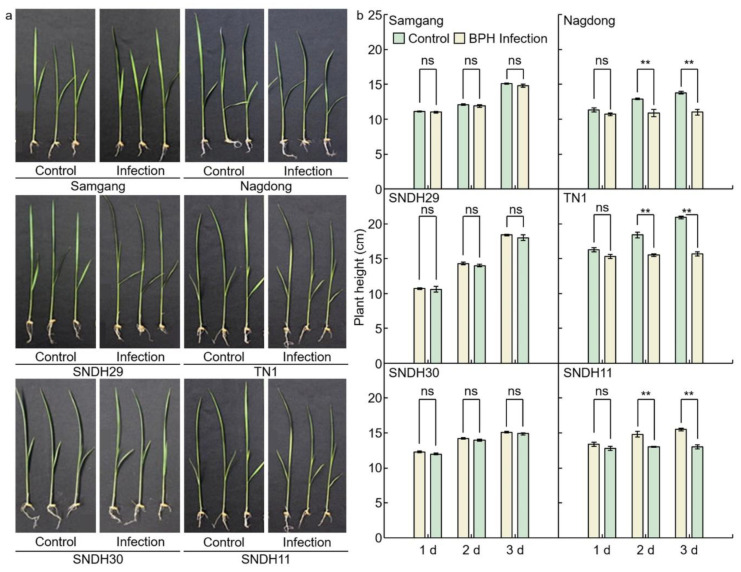
Comparison of phenotypes that are affected after 1, 2, and 3 days of BPH infected in Samgang, Nagdong, TN1, SNDH29, SNDH30 and SNDH11. (**a**) The plant heights were investigated for each group, namely the control and the group infected with BPH, and representative plants of each group in the repeated experiment are shown. (**b**) The plant height over time after being infected with BPH. Control: not infected with BPH; Infection: infected with BPH. 1 d: One day after BPH infection; 2 d: two days after BPH infection; 3 d: three days after BPH infection. Data are shown as the mean ± SD (*n* = 5). ** indicates a significant difference at *p* < 0.01. ns indicates not significant.

**Figure 2 ijms-23-01540-f002:**
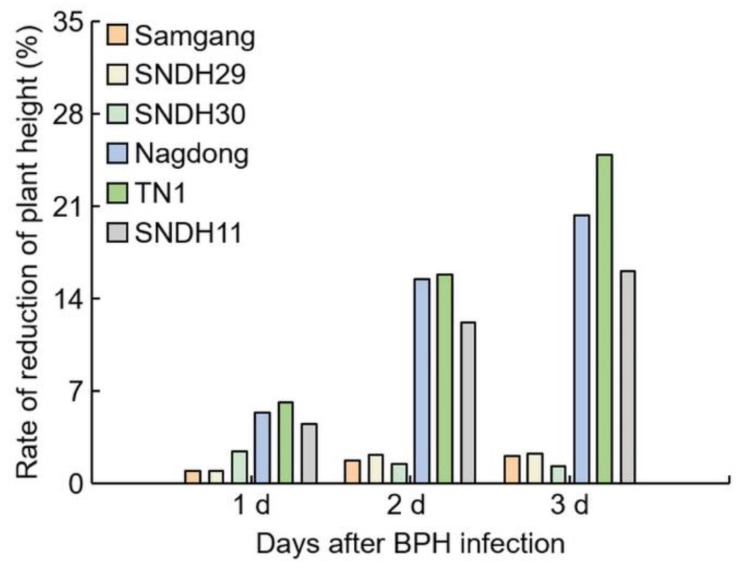
Analysis of the reduction rate of plant growth for 3 days after BPH infection. 1 d: One day after BPH infection; 2 d: two days after BPH infection; 3 d: three days after BPH infection. Bars are shown as the value of the rate of reduction in plant height (*n* = 5).

**Figure 3 ijms-23-01540-f003:**
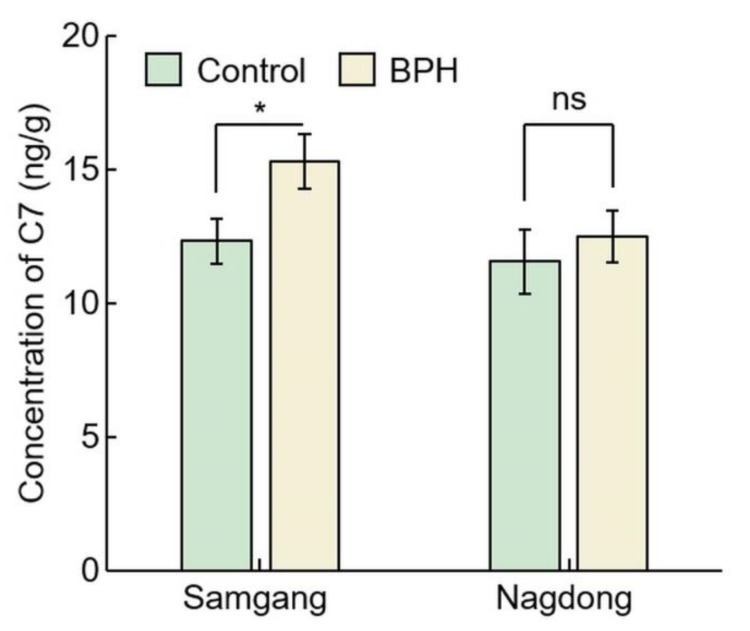
Analysis of concentration of C7 after BPH infection in Samgang and Nagdong. Control: BPH non-infection. BPH: BPH infection. Samgang: BPH-resistant cultivar. Nagdong: BPH-susceptible cultivar. Data are shown as the mean ± SD (*n* = 5). * indicates a significant difference at *p* < 0.05. ns indicates not significant.

**Figure 4 ijms-23-01540-f004:**
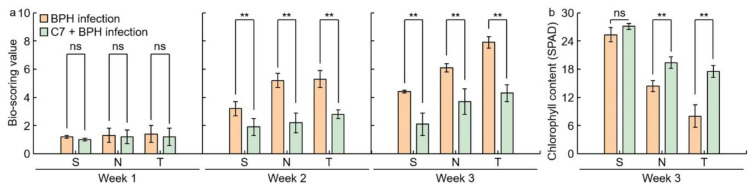
Analysis of C7 efficacy for BPH resistance by cultivars after BPH infection. (**a**) Bio-scoring value was assigned based on the damage level that appears after BPH infection. (**b**) Chlorophyll contents were investigated 3 weeks after BPH infection. S: Samgang; N: Nagdong; T: TN1. Data are shown as the mean ± SD (*n* = 5). ** indicates a significant difference at *p* < 0.01. ns indicates not significant.

**Figure 5 ijms-23-01540-f005:**
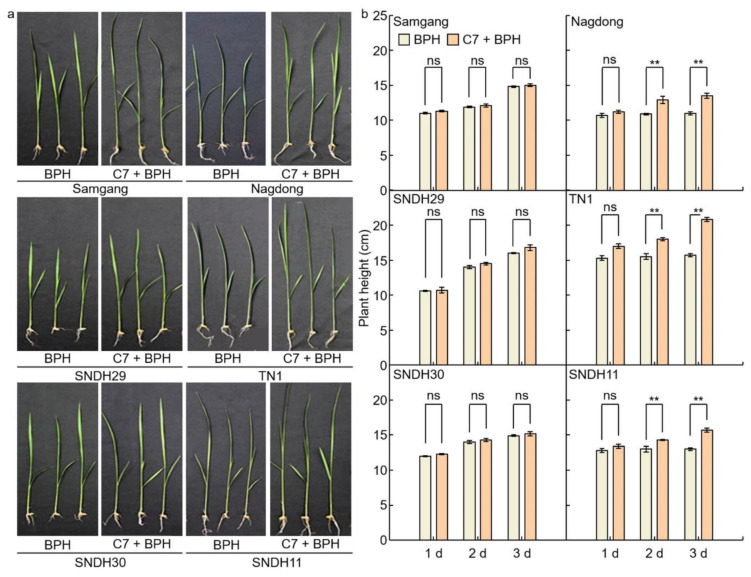
Phenotype comparison for C7 bio-efficacy analysis for BPH in rice. (**a**) The plant height at 3 days after C7 treatment and BPH infection. (**b**) The plant height was investigated over 3 days after C7 treatment and BPH infection. BPH: BPH infection in Samgang, Nagdong, TN1, SNDH29, SNDH30 and SNDH11, C7 + BPH: C7 treatment and BPH infection in Samgang, Nagdong, TN1, SNDH29, SNDH30 and SNDH11. 1 d: One day after BPH infection; 2 d: two days after BPH infection; 3 d: three days after BPH infection. Data are shown as the mean ± SD (*n* = 5). ** indicates a significant difference at *p* < 0.01. ns indicates not significant.

**Figure 6 ijms-23-01540-f006:**
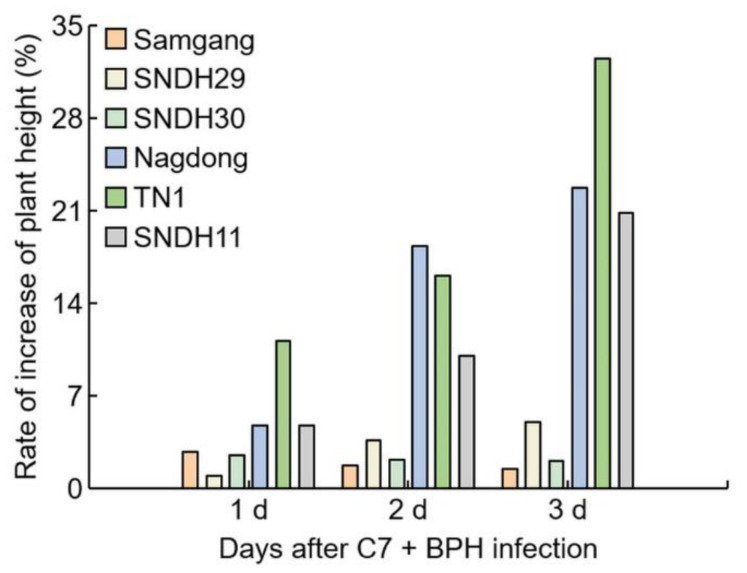
Analysis of the rate of plant growth increase after C7 treatment in plants infected with BPH. 1 d: One day after BPH infection; 2 d: two days after BPH infection; 3 d: three days after BPH infection. Bars are shown as the value of the rate of increase in plant height (*n* = 5).

**Figure 7 ijms-23-01540-f007:**
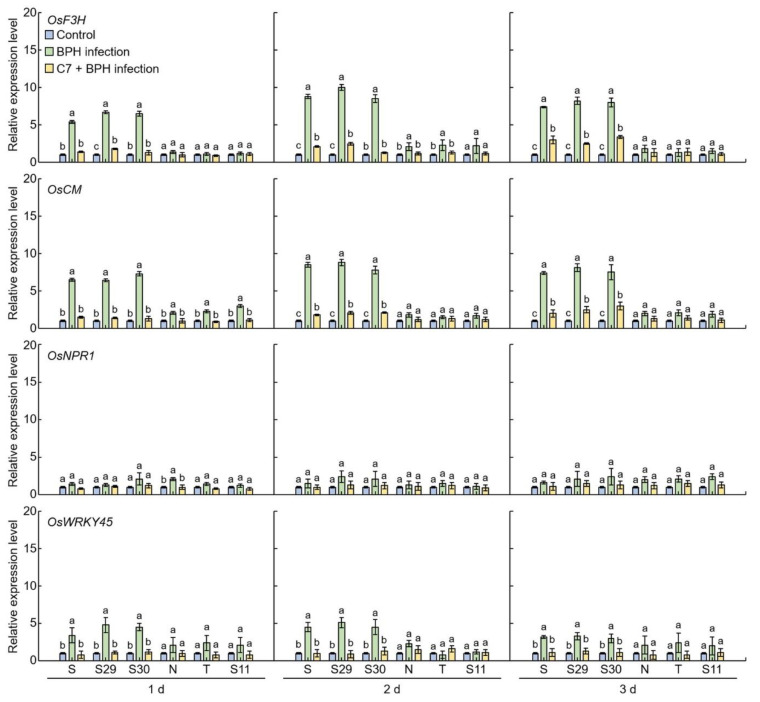
Expression levels of the genes involved in plant defense in the leaves of rice. 1 d: One day after BPH infection; 2 d: two days after BPH infection; 3 d: three days after BPH infection. S: Samgang; S29: SNDH29; S30: SNDH30; N: Nagdong; T: TN1; S11: SNDH11. Bars represent means ± SD (*n* = 5). Mean denoted by the same letter are not significantly different (*p* < 0.05) as evaluated by Duncan’s multiple range test (DMRT).

**Table 1 ijms-23-01540-t001:** Analysis of C7 Concentration After BPH Infection in Samgang and Nagdong.

Cultivar	Concentration of C7 (ng/g)		*p* Value	Rate of Increase (%)
	Control	BPH		
Samgang	12.32 ± 0.85	15.30 ± 1.01	0.0173 *	24.24 ± 1.79
Nagdong	11.54 ± 1.21	12.48 ± 0.96	0.3434 ns	08.41 ± 6.41

Control: BPH non-infection. BPH: BPH infection. Values are shown as the mean ± SD (*n* = 5). * indicates a significant difference at *p* < 0.05. ns indicates not significant.

## Data Availability

Not applicable.

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
