# Peer review of "Bio-Efficacy of Chrysoeriol7, a Natural Chemical and Repellent, against Brown Planthopper in Rice"

_ijms, 2022, doi:10.3390/ijms23031540_

Round 1

Reviewer 1 Report

The manuscript "Bio-efficacy of Chrysoeriol7, a Natural Chemical and Repellent, against Brown Planthopper in Rice" is well conceived, designed and executed, and the methods detailed and thorough. However, the text is not well written (especially the new parts added in the revised version), and in several parts, there are sentences hard to read. Please rewrite (e.g., see lines 49-52, 108-121). The whole manuscript could use a tough edit to really reduce it down to a cleaner, shorter, more concise read.

Moreover, I suggest the Authors to use the terms "infection" and "infested" in a proper way. Infected cannot be used as a synonymous of infested!

Moreover, I list some other comments below, to be addressed before this paper is ready for publication.

Line 15: I suggest to include the scientific name of the target species in the abstract or in the keywords at least to improve the relevancy of the manuscript to readers.

Line 23: change to "... chrysoeriol7 (C7), a secondary...". 

Lines 50, 52: remove the hyphens.

Line: 80: the taxonomic rank is wrong. Please correct.

Line 85: The genus for a scientific name needs to be abbreviated when mentioned again.

Lines 86, 88, etc.: Add the authorship and taxonomic rank for all the insect species reported in the text.

Figure 3: since ns (not significant) is reported in this figure, the Authors should do the same in the Figures 1,4 and 5, as well.

Line 149: revise the English of the caption.

Author Response

Response to Reviewer 1 Comments

Point 1: The manuscript "Bio-efficacy of Chrysoeriol7, a Natural Chemical and Repellent, against Brown Planthopper in Rice" is well conceived, designed and executed, and the methods detailed and thorough. However, the text is not well written (especially the new parts added in the revised version), and in several parts, there are sentences hard to read. Please rewrite (e.g., see lines 49-52, 108-121). The whole manuscript could use a tough edit to really reduce it down to a cleaner, shorter, more concise read.

Response 1: I am grateful for your accurate notation. Lines 49-52 mean that with synthetic pesticides, the speciation that is resistant to synthetic pesticides emerged. Lines 49-52 have been modified to “In fact, research has been reported in which cause the speciation that has resistance to the synthetic pesticides, due to the control of BPH with synthetic pesticides. The main components were such as organophosphates, carbamates, pyrethroids, neonicotinoids, insect growth regulators, and henylpyrazole.”. Lines 108-121 analyzed phenotypic changes after Brown planthopper (BPH) infection in rice. The BPH resistant populations (Samgang, SNDH29, and SNDH30) had no significant growth disorders between the BPH non-infection group and the BPH infection group. In BPH-susceptible populations (Nagdong, TN1, and SNDH11), the BPH infection group had a negative effect on growth, thus resulting in shorter plant heights compared to the control (Figure 1). Lines 108-121 have been modified to “To identify BPH resistance, plant heights were measured on days 1, 2, and 3 after BPH infection to the Samgang, Nagdong, TN1, SNDH29, SNDH30, and SNDH11 (Figure 1). Figure 1a shows representative plants of each group in the repeated experiment. Samgang, SNDH29, and SNDH30 were resistant populations to BPH. Nagdong, TN1, and SNDH11 were susceptible populations to BPH. BPH-resistant populations had no significant difference between the control and BPH infection group. But BPH-susceptible population had a negative effect on growth than the control. The plant heights of the BPH-susceptible population were short from 2 days after BPH infection. In the BPH-susceptible population, the degree of growth disorder due to BPH infection increased over time.”. The newly written introduction and result section have become interesting and easy for readers to understand. Also improved the quality of the manuscript. The modified parts were remarked in red.

Point 2: Moreover, I suggest the Authors to use the terms "infection" and "infested" in a proper way. Infected cannot be used as a synonymous of infested!

Response 2: I am grateful for your accurate notation. Rice damaged by BPH was referred to as infected rice [1,2]. In this manuscript, we analyzed the phenotype and gene expression changes resulting from the BPH infection in rice. Therefore, ‘infection’ and 'infected' were used as appropriate methods in the manuscript. The newly written manuscript has become interesting and easy for readers to understand. Also improved the quality of the manuscript. The revised parts were remarked in red.

  1. Xu, H.X.; He, X.C.; Zheng, X.S.; Yang, Y.J.; Zhang, J. feng; Lu, Z.X. Effects of SRBSDV-infected rice plants on the fitness of vector and non-vector rice planthoppers. J. Asia. Pac. Entomol. 2016, 19, 707–710, doi:10.1016/j.aspen.2016.06.016.
  2. Sun, Z.; Liu, Z.; Zhou, W.; Jin, H.; Liu, H.; Zhou, A.; Zhang, A.; Wang, M.Q. Temporal interactions of plant - insect - predator after infection of bacterial pathogen on rice plants. Sci. Rep. 2016, 6, 1–12, doi:10.1038/srep26043.

Point 3: Line 15: I suggest to include the scientific name of the target species in the abstract or in the keywords at least to improve the relevancy of the manuscript to readers.

Response 3: I am grateful for your accurate notation. To make it easier for readers to understand target species, we added the scientific name of BPH at Abstracts and Keywords. In the first sentence of Abstract, “Brown planthopper (BPH)” has been modified to “Brown planthopper (BPH, Nilaparvata lugens Stal.)”. The newly written abstract has become interesting and easy for readers to understand. Also improved the quality of the manuscript. The revised parts were remarked in red.

Point 4: Line 23: change to "... chrysoeriol7 (C7), a secondary...".

Response 4: I am grateful for your accurate notation. Line 23, “chrysoeriol7, a secondary metabolite” has been modified to “chrysoeriol7 (C7), a secondary metabolite”. The newly written abstract has become interesting and easy for readers to understand. Also improved the quality of the manuscript. The revised parts were remarked in red.

Point 5: Lines 50, 52: remove the hyphens.

Response 5: I am grateful for your accurate notation. All the hyphens on lines 50 and 52 in the manuscript were removed. The newly written introduction has become interesting and easy for readers to understand. Also improved the quality of the manuscript. The revised parts were remarked in red.

Point 6: Line: 80: the taxonomic rank is wrong. Please correct.

Response 6: I am grateful for your accurate notation. The taxonomic rank of Anticarsia gemmatalis was Anticarsia gemmatalis (Hübner) (Lepidoptera: Noctuidae) [3,4]. In the introduction section, the taxonomic rack of Anticarsia gemmatalis was rewritten correctly. The newly written introduction has become interesting and easy for readers to understand. Also improved the quality of the manuscript. The revised parts were remarked in red.

  1. Piubelli, G.C.; Hoffmann-Campo, C.B.; Moscardi, F.; Miyakubo, S.H.; Neves De Oliveira, M.C. Are chemical compounds important for soybean resistance to Anticarsia gemmatalis? J. Chem. Ecol. 2005, 31, 1509–1525, doi:10.1007/s10886-005-5794-z.
  2. Hoffmann-Campo, C.B.; Neto, J.A.R.; De Oliveira, M.C.N.; Oliveira, L.J. Detrimental effect of rutin on Anticarsia gemmatalis. Pesqui. Agropecu. Bras. 2006, 41, 1453–1459, doi:10.1590/S0100-204X2006001000001.

Point 7: Line 85: The genus for a scientific name needs to be abbreviated when mentioned again.

Response 7: I am grateful for your accurate notation. Line 85, to make it easier for readers to understand, the scientific names that have been mentioned again modified to the abbreviation. “Arabidopsis thaliana” has been modified to “Arabidopsis”. The newly written introduction has become interesting and easy for readers to understand. Also improved the quality of the manuscript. The revised parts were remarked in red.

Point 8: Lines 86, 88, etc.: Add the authorship and taxonomic rank for all the insect species reported in the text.

Response 8: I am grateful for your accurate notation. To make it easier for readers to understand, we added the authorship and taxonomic rank to all the insect species written in the manuscript. Pieris brassicae (Lepidoptera: Pieridae) [5], Spodoptera litura (Lepidoptera: Noctuidae) [6] were rewritten correctly. The newly written introduction has become interesting and easy for readers to understand. Also improved the quality of the manuscript. The revised parts were remarked in red.

  1. ROTHSCHILD, M.L.M.S. Assessment of egg load by Pieris brassicae (Lepidoptera: Pieridae). Nature 1977, 266, 352–355.
  2. Armes, N.J.; Wightman, J.A.; Jadhav, D.R.; Rao, G.V.R. Status of insecticide resistance in Spodoptera litura in Andhra Pradesh, India. Pestic. Sci. 1997, 50, 240–248, doi:10.1002/(SICI)1096-9063(199707)50:3<240::AID-PS579>3.0.CO;2-9.

Point 9: Figure 3: since ns (not significant) is reported in this figure, the Authors should do the same in the Figures 1,4 and 5, as well.

Response 9: I am grateful for your accurate notation. To make it easier for readers to understand, we added the ns (indicates not significant) to Figures 1, 4, and 5. The newly written results have become interesting and easy for readers to understand. Also improved the quality of the manuscript. The revised parts were remarked in red.

Point 10: Line 149: revise the English of the caption.

Response 10: I am grateful for your accurate notation To easily understand the sentence for readers, the English of the caption of Table 1 has been modified to “Analysis of C7 concentration after BPH infection in Samgang and Nagdong”. The newly written result has become interesting and easy for readers to understand. Also improved the quality of the manuscript. The revised parts were remarked in red.

Reviewer 2 Report

This is a revised version. It seems that the authors addressed the concerns raised by the reviewers and editor. I have only a minor concern statistical analysis. Regarding Figure 7, were these data subjected to t-test? If so, t-test seems not to be adequate because three groups are used for comparison.

Author Response

Response to Reviewer 2 Comments

Point 1: This is a revised version. It seems that the authors addressed the concerns raised by the reviewers and editor. I have only a minor concern statistical analysis. Regarding Figure 7, were these data subjected to t-test? If so, t-test seems not to be adequate because three groups are used for comparison.

Response 1: I am grateful for your accurate notation. Figure 7 analyzed the relative expression level of each gene (OsF3H, OsCM, OsNPR1, and OsWRKY45) in three groups which are control, BPH infection, and C7+BPH infection. One-way ANOVA was used to determine statistical analysis between mean differences of 3 groups (P < 0.05), followed by Duncan’s multiple range test (DMRT). The statistical analysis method was described in detail to "4.7. Statistical Analysis". Figure 7 and Figure legends are newly represented by applied to one-way ANOVA. Figure 7 indicated the mean difference of each group easily and accurately. The newly represented Figure 7 has become interesting and easy for readers to understand. Also improved the quality of the manuscript. The revised parts were remarked in red.

Round 2

Reviewer 1 Report

I thank the authors for their very detailed replies to comments on the submitted version. They have done a very good revision from their earlier version, and in particular all the comments and suggestions have been addressed. I have no further comments to add.

Author Response

Response to Reviewer 1 Comments

Point 1: I thank the authors for their very detailed replies to comments on the submitted version. They have done a very good revision from their earlier version, and in particular all the comments and suggestions have been addressed. I have no further comments to add.

Response 1: I am grateful for your accurate notation. The newly written manuscript has become interesting and easy for readers to understand. Also improved the quality of the manuscript.
